# Septic Cardiomyopathy: From Pathophysiology to the Clinical Setting

**DOI:** 10.3390/cells11182833

**Published:** 2022-09-11

**Authors:** Federico Carbone, Luca Liberale, Alberto Preda, Thomas Hellmut Schindler, Fabrizio Montecucco

**Affiliations:** 1First Clinic of Internal Medicine, Department of Internal Medicine, University of Genoa, 16132 Genoa, Italy; 2IRCCS Ospedale Policlinico San Martino, Genoa-Italian Cardiovascular Network, 16132 Genoa, Italy; 3Vita-Salute San Raffaele University, 20132 Milan, Italy; 4Mallinckrodt Institute of Radiology, Division of Nuclear Medicine, School of Medicine, Washington University, Saint Louis, MO 63110, USA

**Keywords:** sepsis, septic shock, cardiomyopathy, inflammation

## Abstract

The onset of cardiomyopathy is a common feature in sepsis, with relevant effects on its pathophysiology and clinical care. Septic cardiomyopathy is characterized by reduced left ventricular (LV) contractility eventually associated with LV dilatation with or without right ventricle failure. Unfortunately, such a wide range of ultrasonographic findings does not reflect a deep comprehension of sepsis-induced cardiomyopathy, but rather a lack of consensus about its definition. Several echocardiographic parameters intrinsically depend on loading conditions (both preload and afterload) so that it may be challenging to discriminate which is primitive and which is induced by hemodynamic perturbances. Here, we explore the state of the art in sepsis-related cardiomyopathy. We focus on the shortcomings in its definition and point out how cardiac performance dynamically changes in response to different hemodynamic clusters. A special attention is also given to update the knowledge about molecular mechanisms leading to myocardial dysfunction and that recall those of myocardial hibernation. Ultimately, the aim of this review is to highlight the unsolved issue in the field of sepsis-induced cardiomyopathy as their implementation would lead to improve risk stratification and clinical care.

## 1. Introduction

Since 2016, sepsis has been defined as life-threatening organ dysfunction caused by a dysregulated host response to infection [1]. Such a paradigm shift emphasizes the supremacy of the non-homeostatic host response in infection pathophysiology. Septic patients may be now further categorized in different multidimensional phenotypes: at least four have been identified, not homologous with traditional patient clustering, but strongly correlated with patterns of host immune response and clinical outcomes [2]. When enduring, an inflammatory state may lead to persistent organ injury, and long-term mortality—even by several months to years—in elderly with highly comorbid conditions [3]. Meanwhile, a growing body of evidence is challenging the traditional view on septic cardiomyopathy. Historically assumed as adaptive response to intravascular hypovolemia, cardiac performance actually changes dynamically in accordance with different hemodynamic clusters [4]. Myocardial dysfunction is commonly found in sepsis and outlined by cellular abnormalities, circulating mediators and instrumental parameters. However, whether the degree of myocardial dysfunction is related to systemic organ failure and the extent to which sepsis-induced cardiomyopathy contributes to the outcome remains uncertain and somehow contradictory. This review article aims at summarizing the definition, epidemiology, pathophysiology, and diagnosis of septic cardiomyopathy. A special focus will be paid at identifying and addressing major clinical shortcomings.

## 2. Definition and Epidemiological Considerations

### 2.1. Challenges in Defining Septic Cardiomyopathy

Septic cardiomyopathy was first described in 1921—before the advent of antibiotics—by E. Romberg, in his Lehrbuch der Krankheiten des Herzens und der Blutgefäße (Textbook of Diseases of the Heart and Blood Vessels) as septische akute Myokarditis (septic acute myocarditis). It may be broadly defined as an acute cardiac dysfunction unrelated to ischemia that manifests in different ways: arrhythmias, left and/or right ventricular impairment during systole or diastole, with or without reduction in cardiac output [5,6,7] (Table 1).

Progress in cardiac imaging now makes available several parameters of cardiac performance—historically limited to ejection fraction assessment—but they have paradoxically increased the uncertainty on how to precisely define septic cardiomyopathy. Further issues challenging a global consensus on septic cardiomyopathy include: (i) the potential role of cardiovascular background as premorbid cardiac status; (ii) the need of assessing cardiac function in a setting of highly variable conditions, as determined by inotrope use, autonomic dysregulation of heart rate/arrhythmias, fluid challenge, rise in lactate and reduction in mixed venous oxygen saturation (Figure 1) [5,6,7]. Differentiating septic cardiomyopathy from Takotsubo cardiomyopathy is another relevant clinical issue. Although sepsis and septic shock may trigger Takotsubo cardiomyopathy, the underlying myocardial alterations are different. Furthermore, myocardial dysfunction in Takotsubo cardiomyopathy typically has a regional distribution, mainly an apical ballooning mimicking acute coronary syndrome. Yet, a right ventricular dysfunction cannot be ruled out. Furthermore, Takotsubo cardiomyopathy shows a normalization within days or weeks, although patients may suffer a worse prognosis [8,9,10,11,12,13,14]. The extent to which septic cardiomyopathy overlaps with myocarditis is also difficult to discriminate. Although they share histological markers of stress-induced cardiotoxicity (e.g., interstitial inflammatory infiltrate and edema in conjunction with myocytolysis, necrosis, and interstitial fibrosis), septic cardiomyopathy rarely presents with severe fulminant LV (left ventricular) dysfunction and is generally associated with full recovery of systolic performance [15]. Cardiac function during septic cardiomyopathy should be ultimately conceived as a dynamic assessment over time, not limited to echocardiographic parameters but rather considering the complex hemodynamic context. Gathering clinical, laboratory, and echocardiographic parameters, Geri and colleagues have recently identified five different cardiovascular phenotypes of septic cardiomyopathy (Table 1): (i) no evidence of any cardiac dysfunction; (ii) with LV systolic dysfunction; (iii) hyperkinetic profile, as defined by a preserved—or even supernormal LV systolic function—and an elevated aortic velocity time integral; (iv) right ventricle (RV) failure; (v) persistent hypovolemia [7]. Data-driven machine learning approaches are now unrevealing the importance of continuous monitoring cardiac function to define the correct hemodynamic phenotype [16]. Furthermore, the recent recognition of sepsis phenotypes may provide additional information—or even complicate—the definition and classification of sepsis cardiomyopathy [2]. The analysis of about thirty candidate variables across 1.3 million of patients from three large randomized clinical trials allowed identifying different patterns of organ dysfunction. Especially in the so-called δ phenotype—as compared with the α, β, and γ ones—the cardiovascular involvement was prevalent and associated with increase in inflammatory markers and in abnormal coagulation. Not surprisingly, this phenotype with prevalent cardiovascular involvement was characterized by significantly higher mortality rates. Although δ phenotype is well represented by the classic SOFA score, its characterization deserves more investigations that would also improve definition, characterization, pathophysiology and, hopefully, risk of sepsis cardiomyopathy.

### 2.2. Epidemiology of Septic Cardiomyopathy

Studies on septic cardiomyopathy report a broad range of prevalence ranging from 10% to 70%. Lacking a broad consensus on definition, the epidemiology of septic cardiomyopathy remains elusive. Such a variability reflects the definition used in study designs and the burden of confounding factors: male sex, age, high lactate levels at admission and pre-existing cardiac performance [5]. Long-term effects of septic cardiomyopathy are also poorly addressed but the prevalence of mortality seems to be higher [17] and characterized by a U-shaped non-linear distribution [18]. The timeline raises further issues. The relatively few studies published before and after the Third International Consensus Definitions for Sepsis and Septic Shock (Sepsis-3) are potentially biased in patient selection [17,19,20]. Most of the them are limited to the setting of intensive care unit (ICU), where patients are somehow selected and largely differ from those admitted in internal medicine wards in terms of comorbidities, standards of care and outcome. Cumulative rate of sepsis admitted to the ICU is indeed quite low (44 vs. 367 cases per 100,000 adult/year) [21] and patients are generally younger and with better baseline myocardial performance. Of note, the risk of cardiomyopathy seems to be higher in ICU, as older comorbid patients admitted to internal medicine wards are likely to die before developing cardiac dysfunction. Based on the above considerations, a consensus definition of septic cardiomyopathy is claimed as a milestone for future studies.

## 3. Pathophysiology of Septic Cardiomyopathy

Endothelial, metabolic, and immune response abnormalities are generally involved in the pathogenesis of ventricular dysfunction and arrhythmias during sepsis, whereas the potential role of myocardial ischemia seems limited. Impaired blood flow autoregulation in coronary microcirculation and altered metabolism of lactate, free fatty acid, and glucose likely play a leading role. Septic myocytes may indeed switch their primary energy substrate from free fatty acids to glucose with detrimental effects on contractility, similarly to that observed during post-ischemic myocardial hibernation [22]. However, the transition point from a protective to an adverse and maladaptive process is missing the humoral response in sepsis (Figure 2).

### 3.1. Inflammatory Pathways and Cardiomyocyte Dysfunction

The hyperinflammatory—not counterbalanced—response is historically reported as a paradigm of sepsis. Early clinical trials then focused on blunting inflammatory response by common anti-inflammatory drugs such as glucocorticoids, non-steroidal anti-inflammatory drugs or target therapy, with very limited results [23]. The extreme variability in pro-/anti-inflammatory balance and the wide number of molecular pathways may explain such disheartening results [24,25]. Ancestral signals released by pathogens (pathogen-associated molecular patterns, PAMPs) or by damaged host tissues (damage-associated molecular patterns, DAMPs) and receptors such as Toll-like receptors (TLRs) trigger multiple intracellular pathways, including the activation of nuclear factor-kB (NF-kB) and mitogen-activated protein kinase (MAPK) [26,27,28]. The DAMPs with a recognized role in the pathophysiology of septic cardiomyopathy include: heparan sulphate, which increases intranuclear transcription of pro-inflammatory cytokines and vascular permeability [29]; high mobility group protein B1, which induces loss of calcium from sarcoplasmic reticulum [30]; and histones, which are able to interfere with the production of cellular ATP by reducing mitochondrial membrane potential and causing LV dysfunction and arrhythmias [31]. Among PAMPs, bacterial lipopolysaccharide (LPS) partially mimics hemodynamic effects of septic shock when administered to animal models or humans [32,33]. Detrimental effects of endotoxemia on cardiac contractility are largely due to mitochondrial dysfunction that leads to abnormal calcium handling, disruption of ATP synthesis, endothelium reticulum stress, and autophagy. Furthermore, LPS leads to electrophysiological dysfunctions through a both a direct and cytokine-mediated effect on sodium current kinetics and non-selective cation channel transient receptor potential vanilloid 1 (TRPV1) [34,35,36,37]. Furthermore, LPS and its downstream mediator TNF-α, IL-1β, IL-6, and C5a and ROS [38,39] critically alter calcium (Ca^++^) homeostasis by blunting the amplitude of intracellular currents and concentrations. Dysfunction of intracellular calcium transporters and decreased calcium sensitivity in cardiac myofilaments (due to phosphorylation of inhibitory troponin I) are cornerstones of the impaired excitation-contraction coupling. Ultimately, endotoxemia impairs sarcolemma diastolic Ca^++^ extrusion with consequent overload [40,41,42,43] that determines systolic and diastolic dysfunction during sepsis [44,45]. The characteristic delayed reversibility of septic cardiomyopathy might be explained by the synthesis of new myofilaments to replace the previously phosphorylated (and therefore inactive) ones.

Signals regulated by the TLRs expressed on myocyte surface (TLR2, TRL3, TLR4 and TLR9) lead to transcription of several pro-inflammatory cardiac depressive factors and cytokines such as interleukin (IL)-1, IL-6, tumor necrosis factor (TNF)-α and complement anaphylatoxin C5a [46,47]. LV ejection fraction (LVEF) was found higher in septic mouse with TRL3 and TLR9 gene deletion, suggesting their role in the mechanism of sepsis-induced cardiac dysfunction [48]. Likewise, TLR4 regulates oxidative stress in ryanodine receptor 2 involved in the storage of calcium within sarcoplasmic reticulum of cardiomyocytes [49], and its inhibition showed protective effects [50]. C5a is so far the only complement factor with reported direct myocardial depressive effect, despite several cardiac enzymes related to cardiac dysfunction during sepsis, such as serca2, NCX and Na+/K+-atpase, which are complement receptor dependent [51]. Finally, adhesion molecules take part in homing mechanisms and are upregulated during inflammatory response. In murine coronary endothelium and cardiomyocytes, specific adhesion molecules were upregulated after LPS and TNF-α infusion. Moreover, the use of antibody blocking these molecules such as anti-Intercellular adhesion molecule-1 (ICAM-1) and vascular adhesion molecule-1 (VCAM-1) showed to prevent myocardial neutrophil accumulation and cardiac dysfunction in animal models of sepsis [52]. On the contrary, neutrophils depletion did not protect against myocardial dysfunction during sepsis, suggesting their lesser cardiotoxic potential.

Nitric oxide (NO) is synthesized by nitric oxide synthase (NOS) in different cells of the CV system including cardiac myocytes, and it exerts important roles in maintaining tissue homeostasis by reducing oxidative stress. Indeed, via the cGMP pathway, NO regulates vascular tone, has an antioxidant effect, inhibits leukocyte and platelet adhesion to the endothelium, and increases myocardial contractility [53]. Inflammation and oxidative stress show deep interplay, fueling each other and facilitating the onset of deleterious vicious circles in different diseases including sepsis. Differently from its endothelial and neuronal isoforms, inducible NOS (iNOS) is not constitutively active and when highly expressed is a major responsible for vasodilatation and hypotension in shock [54]. Indeed, iNOS can produce large amount of NO when an inflammatory response occurs [55]. Sepsis leads to overexpression of iNOS not only in immune cells but also in the myocardium [56]. Such increased expression has detrimental effects on contractile function of cardiomyocytes, in part through the paradoxical induction of reactive oxygen species (ROS), i.e., peroxynitrite [57], in part through the down-regulation of adrenaline receptors and decreased sensitivity to calcium [29]. Confirming the important role played by iNOS in myocardial dysfunction during sepsis, several non-specific anti-NOS drugs such as melatonin and methylene blue showed beneficial effects in terms of cardiovascular function and prognosis in preclinical studies [57,58].

### 3.2. Adrenergic System

Increased activation of the sympathetic system is a major compensatory reaction to septic-related vasoplegia. During sepsis, β-adrenergic receptors (βARs) are downregulated, and the responsiveness to catecholamines is reduced in the whole cardiovascular system [59,60]. When persistent, this adaptive response become maladaptive [61]. In myocardial tissue, excessive stimulation of cardiac βARs suppresses their expression and leads to an inhibitory response with reverse of adrenergic G protein coupling [62], intracellular calcium overload [63], increased production of ROS [64], disruption of membrane potential via the inhibition of Na+/K+-ATPase pump [65], and induction of apoptosis [66]. A relevant role is played by Cardiac G Protein-Coupled Receptor Kinase 2 (GRK2). They are the major negative regulators of βAR pro-contractile signal in sepsis-induced myocardial dysfunction [67] and other different cardiac diseases including chronic heart failure and Takotsubo syndrome [68].

### 3.3. Microvascular Dysfunction and Vasoactive Peptides

Activation of endothelial cells in consequence of an infective stimulus leads to the synthesis of inflammatory cytokines, increased expression of cell adhesion molecules, loss of the barrier function by glycocalyx shedding, edema, apoptosis, hypercoagulative state and vasoplegia [69]. Several endogenous vasoactive peptides generate and maintain endothelial dysregulation. LPS and cytokines up-regulate cyclooxygenase (COX)-2 (inducible form) and then the production of prostanoids from arachidonic acid. This process mainly occurs in inflammatory cells, myofibroblasts, endothelium and even in cardiomyocytes (i.e., in myocardial infarction) [70,71]. Elevated levels of prostanoids such as thromboxane and prostacyclin are associated with coronary microvascular dysfunction [72]. However, clinical trials have failed so far in proving any beneficial effect of pharmacological COX inhibition on coronary microvascular homeostasis in sepsis. Less is known about the potential role of endothelin-1 (ET-1) in sepsis cardiomyopathy, whereas an increase has been reported in chronic cardiac disease (i.e., heart failure) [73]. ET-1 would have a major role in infectious disease, as a promoter of cytokines release, platelet aggregation and vasoreactivity [74]. In single mouse study, cardiac overexpression of ET-1 was associated with the onset of severe sepsis cardiomyopathy, characterized by interstitial infiltration of macrophages and T lymphocytes and increased levels of the pro-inflammatory cytokines TNF-α, INF-γ, IL-1, and IL-6 [75].

### 3.4. Energetic Dysmetabolism

Sepsis is characterized by altered myocardial lipoprotein metabolism and mobilization of triglycerides and free fatty acids to overwhelm the systemic suppression of energy production [76]. Physiologically, about 70% of cardiac ATP is produced via lipid oxidation, while the rest is produced via glucose oxidation. A minor part also derives from the catabolism of lactate and ketone bodies. Under pathological conditions, glucose oxidation becomes the prevalent energetic pathway [77]. During sepsis, a reduction in fatty acid oxidation is not compensated by the increase in glucose catabolism due to altered insulin action, exacerbation of the inhibitory effects of alternative substrates on glycolysis, and blunted glycogen synthesis [78]. Several enzymes involved in intracellular cardiac fatty acid mobilization and oxidation are also inhibited [79,80]. Cardiac transcriptional factors associated with fatty acid oxidation, such as peroxisome proliferator activated receptors (PPARs) are also suppressed in sepsis and their stimulation with PPARs agonists lead to improved survival in septic mice [81]. Mitochondrial dysfunction further contributes to the energetic failure in sepsis cardiomyopathy, thus being a non-negligible cause of reduced outcome [82,83,84]. Inflammation and oxidative stress alter mitochondrial structure determining swelling, cytoplasmic accumulation of denatured protein and lysosomal lesions [85,86]. Such a damage interferes with the respiratory chain [87] with falling in ATP synthesis, release of calcium and pro-apoptotic proteins [88]. The pro-oxidant environment induced by endotoxemia also trigger cardiac mitophagy, a defensive mechanism for the removal of damaged mitochondria [89]. The antioxidant N-acetylcysteine seems to have a protective roles against contractile dysfunction and mitophagy. Even levosimendam in both mice and human beings has beneficial effects due to calcium sensitization and antioxidant properties [90]. Lastly, when the degree of mitochondrial dysfunction is mild, myocardial hibernation may occur. This is an adaptive, self-protective mechanism during which all functions get reduced. Such a downregulation of mitochondrial gene transcription has been reported in sepsis cardiomyopathy [91] and represents a reversible condition that improves together with the resolution of sepsis [84].

## 4. Clinical Diagnosis

A combination of clinical, biochemical and hemodynamic parameters is increasingly described as the best approach to identify and stratify septic cardiomyopathy.

### 4.1. Doppler Echocardiography: From Classic to Innovative Assessment

Echocardiography is considered the first line approach for the global assessment of patients with septic shock. The use of ultrasound in this field has progressed over years but the application of echocardiography for diagnosis/monitoring of septic cardiomyopathy remains challenging. The dependence on loading conditions is a not overcoming intrinsic limitation for several echocardiographic parameters, which rather reflect the degree of fluid resuscitation/vascular leak (i.e., preload) and vasoplegia/response to vasoactive drugs (i.e., afterload) [7]. This is especially true for LVEF, traditionally considered the hallmark of septic cardiomyopathy. Indeed, merely identifying low LVEF (<50%) during sepsis does not have predictive value in these patients [92]. Rather, LVEF would reflect the abovementioned U-shaped non-linear distribution. Indeed even a hyperdynamic LVEF—mostly expression of reduced afterload—is burdened by higher short-term mortality [93]. As partially afterload-independent measures, tissue Doppler imaging (TDI) and speckle-tracking echocardiography (STE) are promising candidates. Unlike the peak systolic velocity of the mitral annulus (referred to as S’) [20], the global longitudinal strain (GLS) is substantially more performing. GLS is a semi-automated algorithm tracking selected region of myocardium during cardiac cycle (more specifically the ratio of the maximal change in the myocardial longitudinal length in systole to the original length in diastole). GLS may detect the impairment of LV systolic function before LVEF declines and worse GLS values are associated with higher mortality in patients with severe sepsis and septic shock [94]. Looking at the future, coupling afterload and cardiac performance might optimize echocardiographic assessment by overcoming this critical bias.

A rise in end-diastolic volume is common in septic shock patients—especially in those receiving large volemic replacement—and likely associated with poor outcome. The rationale for considering diastolic dysfunction in sepsis-induced cardiomyopathy relies on the upstream effect on pulmonary circulation, right heart, and finally tissue edema. However, the mining of diastolic dysfunction in sepsis cardiomyopathy remains to be elucidated. The concept of sepsis-induced cardiomyopathy is originally based on the abnormal LV function, but the risk ratio of patients with concomitant diastolic dysfunction is not yet established. In particular, the TDI-derived velocity of the mitral annulus during early diastole (e’)—considered relatively preload independent—and the ratio with the peak of trans-mitral inflow velocity (E) are substantially associated with mortality [95]. However, pathophysiological and therapeutic implications of diastolic dysfunction are still weak. The assessment of diastolic function is not yet validated for critically ill patients, and data extrapolation is very challenging when the increase in LV end-diastolic pressure is influenced by underlying cardiac performance (frequently unknown) and a variable degree of fluid resuscitation/vascular leak). Furthermore, mitral annulus velocity is influenced by heart rate in hyperdynamic cardiac output.

Right ventricle (RV) dysfunction is increasingly being reported in patients with sepsis. It can be defined as a fractional area change (FAC) less than 35% or tricuspid annulus systolic plane excursion (TAPSE)—the most reproducible measure—less than 16 mm. Systolic RV pressure may also be easily calculated from tricuspid regurgitation flow. RV dysfunction is common in severe sepsis/septic shock. Its prevalence ranges from 40% to 60% at admission but significantly increase when the enrollment lasts up to 72 h. RV dysfunction correlated with longer stay in ICU, and independently associated with higher risk of both early and long-term mortality [96,97,98]. Nevertheless, conflicting results exist and the complex geometry of RV—alongside the tight ventricular interdependence—raises many questions. RV is a thin-walled structure facing off a high-compliance/low-resistance pulmonary circulation. In sepsis, a paradoxical increase in pulmonary vasculature resistance may occur, due to hypoxic vasoconstriction, pulmonary source of infection/acute respiratory distress syndrome (ARDS), hypercapnia, and mechanical ventilation. Fluid overload further increases the preload on RV, leading to isolated RV dysfunction. In addition, LV dysfunction remains the leading cause of RV dysfunction and the association with poor outcome might be rather a “false positive” [99].

### 4.2. Cardiac Performance

The afterload-related cardiac performance (ACP) is a relative old concept firstly introduced in 2011 and defined as [100]:ACP (%) = CO_measured_/CO_predicted_ × 100%

In turn, Cardiac Output (CO)_predicted_ may be calculated as a function of systemic vascular resistance (SVR):CO_predicted_ = β0 × SVR^β1^ (β0 = 394.07, β1 = −0.64)

In preliminary studies, ACP estimation correlates with both APACHE II and SOFA scores, even improving mortality prediction in both early and late stages of sepsis. ACP is now considered as valid prognostic marker in sepsis and septic shock, ACP values lower than 80% are associated with a higher risk of mortality in septic patients [101,102]. When defining septic cardiomyopathy by ACP (<80%), a recent study from Wilhelm and coll. reported that about 50% of patients with septic shock have at least a moderate septic cardiomyopathy, and that the lower the ACP the higher is mortality in this group of patients [103]. While validation in larger studies is needed, the use of a machine learning approach to implement this algorithm might further improve the model’s performance. Diameter and collapsibility of inferior vena cava and B line quantification at lung ultrasound are further intriguing surrogate measures of cardiac performance that deserve future investigations.

### 4.3. Biomarkers

The lack of a consensus definition and the uncertainties on pathophysiology of septic cardiomyopathy makes it difficult to identify and validate biomarkers of disease. Being associated with cardiac injury, troponins and natriuretic peptides rise during sepsis and correlate with the degree of myocardial dysfunction [104]. The mechanisms of troponin release during sepsis remain to be fully elucidated, but inflammation, myocardial wall stress, drug toxicity and kidney dysfunction rather than cell death are likely be implicated. Newly developed echocardiographic parameters, including TDI e STE applied to both left and right ventricles correlated with troponin levels [105], but their association with mortality likely reflects the severity of disease rather than the effective degree of myocardial dysfunction [94].

Brain natriuretic peptides (BNP and NT-proBNP) are secreted by myocardial cells in response to stretch. This would candidate BNPs as surrogate biomarkers of LV dysfunction. A significant association with both systolic and diastolic LV dysfunction was actually—but not unequivocally—demonstrated for BNPs. Preliminary data also link the rise in BNPs with RV dysfunction [106]. However, increased BNP levels during sepsis may be due to different conditions including lung involvement (e.g., source of infection, ARDS, mechanical ventilation), pre-existing cardiac dysfunction, and renal impairment. BNPs also failed to predict the responsiveness to fluid challenge, and cannot be used as a reliable biomarker of volemic status during severe sepsis/septic shock 98. Even concerning mortality, the rise in BNPs is associated with higher risk in meta-analyses but with substantial heterogeneity across the studies [107,108]. As for acute decompensated heart failure, serial measurement of BNPs might improve the performance of these biomarkers and clarify whether the prognostic value is specifically tied with cardiac dysfunction or rather with sepsis severity.

The emerging field of non-coding RNAs (ncRNAs) recently opened new scenarios for modulation of inflammation in sepsis, including sepsis-related cardiomyopathy 101. Among the wide array of ncRNA, a massive amount of experimental data concerns microRNAs (miRNAs) with good potential for clinical translation [109]. Among the growing number of miRNAs with clinical interest for sepsis/induced cardiomyopathy, the best predictors of cardiac injury are miR21-3p and miR-155, but correlations with myocardial dysfunction degree are also reported for miR-135a and miR-494-3p.

The cytokine storm makes it difficult to identify biomarkers of sepsis cardiomyopathy across cytokines/interleukins. A single observational study has reported an association with IL-8 [110]. Something more might be expected from artificial intelligence in the next future. Bioinformatics analyses has the potential for deciphering this complex signals. Tumor necrosis factor, Jak-signal transducer and activator of transcription (STAT), hypoxia-inducible transcription factor-1, and their interactions are increasingly described as crossroads of sepsis cardiomyopathy [111].

## 5. Conclusions and Outlook

The most important goal in the field of septic cardiomyopathy might be its recognition. The lack of a consensus definition led to divergence about its clinical relevance, whereas the advance in echocardiographic technology/algorithm is worryingly—and paradoxically—tangling up the search for a general agreement. Of paramount importance for standardizing the definition of septic cardiomyopathy would be to discriminate whether this is primitive and how it can induce hemodynamic perturbances. In addition, the dynamicity of changes in myocardial function should be considered. There is also growing awareness that mild declines in cardiac performance are rather to be considered as an adaptive response. In line with these missing points, studies outlining treatment goals for sepsis cardiomyopathy are lacking. The “early goal-directed therapy” still remains the current approach, since 2001, although this was not confirmed in later clinical trials (e.g., ProCESS, ARISE, ProMISe) [112,113,114]. As for sepsis, infection control and optimization of hemodynamic parameters are also considered the standard treatment for sepsis cardiomyopathy. Even the assumption of complete recovery in the long-term after the sepsis need to be revised. Long-term outcomes need to be added in study protocols, and this requires adequate infrastructures for monitoring cardiovascular health status after severe sepsis/septic shock. Overall, those unsolved questions have so far limited any therapeutic implications. Implementing those unsolved issues might improve risk stratification and ultimately promote the translation of therapeutic approaches now confined to the experimental field.

## Figures and Tables

**Figure 1 cells-11-02833-f001:**
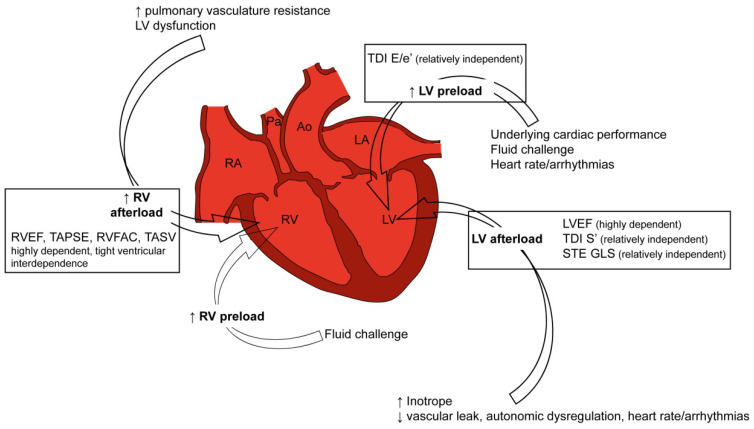
Diagnostic challenges in septic cardiomyopathy are a consequence of its complex pathophysiology. Recent advances in cardiac imaging paradoxically increased the uncertainty on how to define sepsis-induced cardiomyopathy. Underlying myocardial dysfunction, heart rate and the need of acute interventions (i.e., fluid challenge and inotrope use) add further layer of complexity. These issues highlight the importance of a dynamic assessment of cardiac performance over time, not limited to echocardiographic findings but also considering the cardiac performance.

**Figure 2 cells-11-02833-f002:**
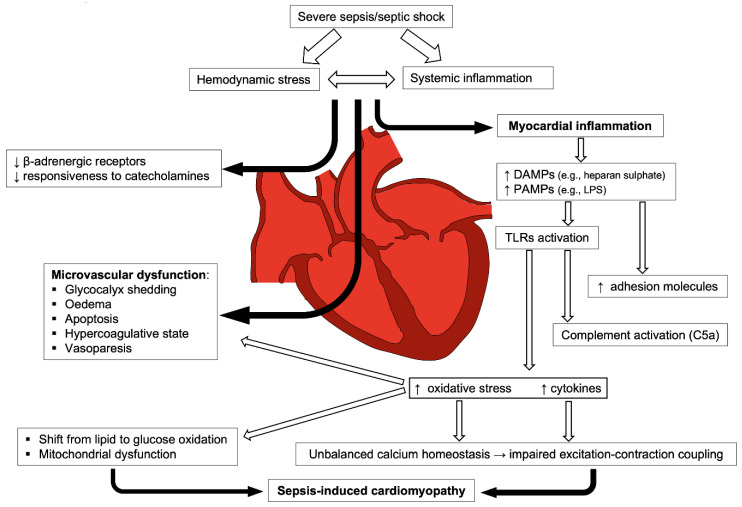
Hemodynamic stress and systemic inflammation exert a synergistic effect on myocardial function impairment during sepsis. Systemic response to sepsis triggers and sustains myocardial dysfunction through several pathways: by impairing the compensatory reaction of sympathetic system with consequent microvascular dysfunction and—In parallel—through theimmune system activation and microvascular dysfunction (black arrows). Those mechanisms deeply impair myocardial tissue through a cascade involving damage- and pathogen-associated molecular pattern (DAMPs and PAMPs, respectively), Toll-like receptors (TLRs), complement and release of cytokines and reactive oxygen species.

**Table 1 cells-11-02833-t001:** Definition of septic cardiomyopathy.

Refs	Phenotypes	Defining Criteria
Diagnostic criteria for sepsis-associated cardiac dysfunction unrelated to ischemia *. Beesley et al., 2018 [5] Martin at al., 2019 [6]	(i) LV dilatation with normal- or low-filling pressure	NA
	(ii) Reduced LV ventricular contractility	LVEF < 40–50%, ↓ LVFAC
	(iii) RV dysfunction of LV (systolic or diastolic) dysfunction with reduce response to fluid challenge	LVEF < 40–50%, ↓ LVFAC, e’ velocity, MPI, Afterload-related cardiac performance, Ventricular arterial decoupling
Cardiovascular phenotypes of septic cardiomyopathy classification performance Geri et al., 2019 [7]	(i) well resuscitated without myocardial dysfunction or fluid responsiveness	–
	(ii) LV systolic dysfunction	LVEF < 40%, LVFAC < 33%, aortic VTI < 14 cm
	(iii) Hyperkinetic profile	Aortic VTI > 20 cm, LVFAC > 58%, heart rate < 106 bpm
	(iv) RV failure	RV/LV EDA > 0.8, sBP < 100 mmHg, dBP < 51 mmHg
	(v) persistent hypovolemia	Aortic VTI < 16 cm, E wave < 67 cm/s, ΔDSVC > 39%

* One or more of these characteristics may define septic cardiomyopathy. LV: left ventricular; LVEF: left ventricular ejection fraction; LVFAC: left ventricular fractional area change; RV: right ventricle; MPI: myocardial performance index; MVG: myocardial velocity gradient; STE: speckle tracking echocardiography; VTI: velocity-time integral; EDA: end-diastolic area; sBP: systolic blood pressure; dBP: diastolic blood pressure; ΔSVC: respiratory variations of the superior vena cava.

## Data Availability

Not applicable.

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
