# Peer review of "Septic Cardiomyopathy: From Pathophysiology to the Clinical Setting"

_cells, 2022, doi:10.3390/cells11182833_

Round 1

Reviewer 1 Report

Carbone et al. present a review article about septic cardiomyopathy. It presents current knowledge and pathomechanisms.

The paper is written and important. However, I have some suggestions

1-The paper should be checked for grammatical and typical errors

2-Please desribe how can we differntiate between myocarditis, septic cardiomyopathy and takotsubo cardiomyopathy.

Of note, Takotsubo cardiomyopathy is also triggered by infections and shows a normalization of ventricular dysfunction within days or weeks. Patients may suffer a worse prognosis. In addition a right ventricular dysfunction is common.

(PMID: 34315238, PMID: 34052189, PMID: 31713085, PMID: 30581088, PMID: 30528096PMID: 28453621)

3-I missed any pathomechanisms on cellular level for ventricular dysfunction and/or for mechanisms of arrhythmias. This should be presented. (PMID: 28592841PMID: 34282193, PMID: 35448095, PMID: 35216067)

4-How can patients with septic cardiomyopathy be managed? what about biomarkers except cardiac biomarkers? any role of Interleukine? cytokine?

Author Response

1-The paper should be checked for grammatical and typical errors. Reply: we apologize for that. We have checked the manuscript to correct grammatical errors and typos.

2-Please describe how can we differentiate between myocarditis, septic cardiomyopathy and takotsubo cardiomyopathy. Of note, Takotsubo cardiomyopathy is also triggered by infections and shows a normalization of ventricular dysfunction within days or weeks. Patients may suffer a worse prognosis. In addition, a right ventricular dysfunction is common (PMID: 34315238, PMID: 34052189, PMID: 31713085, PMID: 30581088, PMID: 30528096, PMID: 28453621). Reply: we thank the reviewer for his/her comment. We have now discussed differential diagnostic challenges between these conditions on page 3, as follows: “…Differentiating septic cardiomyopathy from Takotsubo cardiomyopathy is another relevant clinical issue. Although sepsis and septic shock may trigger Takotsubo cardiomyopathy, the underlying myocardial alterations are different. Furthermore, myocardial dysfunction in Takotsubo cardiomyopathy typically has a regional distribution, mainly an apical ballooning mimicking acute coronary syndrome, but a right ventricular dysfunction cannot be ruled out. Furthermore, Takotsubo cardiomyopathy shows a normalization within days or weeks, although patients may suffer a worse prognosis. (Refs). The extent to which septic cardiomyopathy overlaps with myocarditis is also difficult to discriminate. Although they share histological markers of stress-induced cardiotoxicity (e.g., interstitial inflammatory infiltrate and edema in conjunction with myocytolysis, necrosis, and interstitial fibrosis), septic cardiomyopathy rarely presents with severe fulminant LV (left ventricular) dysfunction and is generally associated with full recovery of systolic performance (Ref)...”.

3-I missed any patho-mechanisms on cellular level for ventricular dysfunction and/or for mechanisms of arrhythmias. This should be presented. (PMID: 28592841, PMID: 34282193, PMID: 35448095, PMID: 35216067). Reply: the reviewer is right. We have now implemented this part on page 5, where the title of paragraph has been modified in “Inflammatory pathways and cardiomyocyte dysfunction”. We have also rewritten the whole paragraph to address the reviewer comments, as follows: “…Detrimental effects of endotoxemia on cardiac contractility are largely due to mitochondrial dysfunction that leads to abnormal calcium handling, disruption of ATP synthesis, endothelium reticulum stress, and autophagy. Furthermore, LPS lead to electrophysiological dysfunctions through a both a direct and cytokine-mediated effect on sodium current kinetics and non-selective cation channel transient receptor potential vanilloid 1 (TRPV1) (Refs). Furthermore, LPS and its downstream mediator TNF-α, IL-1β, IL-6, and C5a and ROS (Refs) critically alter calcium (Ca++) homeostasis by blunting the amplitude of intracellular currents and concentrations. Dysfunction of intracellular calcium transporters and decreased calcium sensitivity in cardiac myofilaments (due to phosphorylation of inhibitory troponin I) are cornerstones of the impaired excitation-contraction coupling. Ultimately, endotoxemia impairs sarcolemma diastolic Ca++ extrusion with consequent overload (Refs) that determines systolic and diastolic dysfunction during sepsis (Refs). The characteristic delayed reversibility of septic cardiomyopathy might be explained by the synthesis of new myofilaments to replace the previously phosphorylated (and therefore inactive) ones.…”.

4-How can patients with septic cardiomyopathy be managed? what about biomarkers except cardiac biomarkers? any role of interleukins? cytokines? Reply: we thank the reviewer for her/his suggestion. We have now implemented this part on page 10 “…The cytokine storm makes difficult identifying biomarkers of sepsis cardiomyopathy across cytokines/interleukins. A single observational study has reported an association with IL-8 (Ref). Something more might be expected from artificial intelligence in the next future. Bioinformatics analyses has the potential for deciphering this complex signals. Tumor necrosis factor, Jak-signal transducer and activator of transcription (STAT), hypoxia-inducible transcription factor-1, and their interactions are increasingly described as crossroads of sepsis cardiomyopathy (Ref)…”, and later on page 11: “…In line with these missing points, studies outlining treatment goals for sepsis cardiomyopathy are lacking. The “early goal-directed therapy” till remains the current approach since 2001, although not confirmed in later clinical trials (e.g., ProCESS, ARISE, ProMISe) (Refs). As for sepsis, infection control and optimization of hemodynamic parameters are considered the standard treatment also for sepsis cardio-myopathy…”.

Reviewer 2 Report

Introduction:

1. Table 1 should be more informative and systematic with numbers for phenotyps to alleviate confusion.

2. It is not quite clear is there any relationship btween different phenotyps and etiology of sepsis. Is there experienced or scientific data on the prevalence of some phenotypes in relationship to their etiology?

3. In chapter Patophysiology of septic cardiomyopthy the subtitels should be more clear such as Response of adrenergic system or Changes in homeostasis in stead of Calcium...

4. In chapter Clinical diagnosis subtitel should be something like "Doppler echocardiography ; traditional and modern paremeters 

definition and criteria for LV diastolic dysfunction is now clear according to the ESC and ACC new guidelines

It will be useful to add the few very important Doppler echo criteria to distiquish level of heart impairement such as: Vena cava inferior diameter and colapsibility, systolic RV pressure, easy calculating from tricuspid regurgitation flow, and pulmonary ultrasound and B lines of appearance and number  of lines. All of those parameters are avaliable when examining patients in bad and have clinical importance. 

There is no data about the inflamatory biomarkers such as c- reactive protein and others ragarding to the patophysiology of septic cardiomyopathy  

Author Response

1. Table 1 should be more informative and systematic with numbers for phenotypes to alleviate confusion. Reply: as suggested we have now added the number for phenotypes and make the Table more informative.

2. It is not quite clear is there any relationship between different phenotypes and etiology of sepsis. Is there experienced or scientific data on the prevalence of some phenotypes in relationship to their etiology? Reply: we thank the reviewer for her/his suggestion. For the best of our knowledge there is no mention in literature on any relationship between the phenotype of sepsis cardiomyopathy, their prevalence and the etiology or sepsis.

3. In chapter Pathophysiology of septic cardiomyopathy the subtitles should be more clear such as Response of adrenergic system or Changes in homeostasis instead of Calcium... Reply: in this revision version of the manuscript this subchapter has been moved in the above one, now renamed: “Inflammatory pathways and cardiomyocyte dysfunction”.

4. In chapter Clinical diagnosis subtitle should be something like "Doppler echocardiography; traditional and modern parameters. Reply: we thank the reviewer for his/her comment. We have now changed the title of the paragraph in: “Doppler echocardiography: from classic to innovative assessment”.

definition and criteria for LV diastolic dysfunction is now clear according to the ESC and ACC new guidelines. Reply: we apologize for this mistake. The sentence has been now rewritten on page 8, as follows: “…However, the mining of diastolic dysfunction in sepsis cardiomyopathy remains to be elucitated...”.

It will be useful to add the few very important Doppler echo criteria to discriminate level of heart impairment such as: Vena cava inferior diameter and collapsibility, systolic RV pressure, easy calculating from tricuspid regurgitation flow, and pulmonary ultrasound and B lines of appearance and number of lines. All of those parameters are available when examining patients in bad and have clinical importance. Reply: we thank the reviewer for these tips. As there are not clinical studies focusing of this measures, we discussed them within the text on page 9 “…Systolic RV pressure may also easily calculated from tricuspid regurgitation flow.”.ì and later on page 9, as follows: “…Diameter and collapsibility of inferior vena cava and B line quantification at lung ultra-sound are further intriguing surrogate measures of cardiac performance that deserve future investigations.”.

There is no data about the inflammatory biomarkers such as c- reactive protein and others regarding to the pathophysiology of septic cardiomyopathy. Reply: the reviewer is right. We have now firstly listed cytokines as downstream signals of endotoxemia on page 6: “…Detrimental effects of endotoxemia on cardiac contractility are largely due to mitochondrial dysfunction that leads to abnormal calcium handling, disruption of ATP synthesis, endothelium reticulum stress, and autophagy. Furthermore, LPS lead to electrophysiological dysfunctions through a both a direct and cytokine-mediated effect on sodium current kinetics and non-selective cation channel transient receptor potential vanilloid 1 (TRPV1) (Refs). Furthermore, LPS and its downstream mediator TNF-α, IL-1β, IL-6, and C5a and ROS (Refs) critically alter calcium (Ca++) homeostasis by blunting the amplitude of intracellular currents and concentrations. Dysfunction of intracellular calcium transporters and decreased calcium sensitivity in cardiac myofilaments (due to phosphorylation of inhibitory troponin I) are cornerstones of the impaired excitation-contraction coupling. Ultimately, endotoxemia impairs sarcolemma diastolic Ca++ extrusion with consequent overload (Refs) that determines systolic and diastolic dysfunction during sepsis (Refs). The characteristic delayed reversibility of septic cardiomyopathy might be explained by the synthesis of new myofilaments to replace the previously phosphorylated (and therefore inactive) ones.…”. Furthermore, we have discussed the clinical relevance of inflammatory biomarkers on page 10 “…The cytokine storm makes difficult identifying biomarkers of sepsis cardiomyopathy across cytokines/interleukins. A single observational study has reported an association with IL-8 (Ref). Something more might be expected from artificial intelligence in the next future. Bioinformatics analyses has the potential for deciphering this complex signals. Tumor necrosis factor, Jak-signal transducer and activator of transcription (STAT), hypoxia-inducible transcription factor-1, and their interactions are increasingly described as crossroads of sepsis cardiomyopathy (Ref)…”.

Round 2

Reviewer 1 Report

Thank you for revising the paper

Author Response

We thank the reviewer for her/his comment